# Locomotion control during curb descent: Bilateral ground reaction variables covary consistently during the double support phase regardless of future foot placement constraints

Chuyi Cui[ID]◉, Ashwini Kulkarni[ID]◉, Shirley Rietdyk◉, Satyajit Ambike[ID]*

Department of Health and Kinesiology, Purdue University, West Lafayette, Indiana, United States of America

◉ These authors contributed equally to this work.
* sambike@purdue.edu

**Data Availability Statement:** All relevant data and code are archived with the Purdue University Research Repository. doi:10.4231/65GZ-K658.

## Abstract

During community ambulation, anticipatory adaptations in gait are key for navigating built, populated and natural environments. It has been argued that some instability in gait can be functionally beneficial in situations demanding high maneuverability, and while the mechanisms utilized to maintain locomotor balance are well understood, relatively less is known about how the control of gait stability changes to facilitate upcoming maneuvers in challenging environments. The double support phase may be important in this regard; since both feet can push off the ground simultaneously, there is greater control authority over the body's movement during this phase. Our goal was to identify how this control authority is exploited to prepare for upcoming maneuvers in challenging environments. We used synergy indices to quantify the degree of coordination between the ground reaction forces and moments under the two feet for stabilizing the resultant force and moment on the body during the double support phase of curb descent. In contrast to our expectations, we observed that the kinetic synergy indices during curb descent were minimally influenced by expected foot targeting maneuvers for the subsequent step. Only the resultant moment in the frontal plane showed reduced stability when targeting was required, but the synergy index was still high, indicating that the resultant moment was stable. Furthermore, the synergy indices indicated that the main function of the ground reaction variables is to maintain stability of whole-body rotations during double support, and this prerogative was minimally influenced by the subsequent foot targeting tasks, likely because the cost of losing balance while descending a curb would be higher than the cost of mis-stepping on a visual target. Our work demonstrates the salience of stabilizing body rotations during curb negotiation and improves our understanding of locomotor control in challenging environments.

**Funding:** SA: Purdue University COVID-19 Research Disruption Fund. https://www.purdue.edu/research/oevprp/funding-and-grant-writing/funding/overview.php CC: Purdue University Department of Health and Kinesiology, Templin Graduate Student research award. https://www.purdue.edu/hhs/hk/graduate/scholarships/grants.html#:~:text=The%20Templin%20Graduate%20Student%20Research,research%20and%2For%20travel%20activities. The funders had no role in study design, data collection and analysis, decision to publish, or preparation of the manuscript.

**Competing interests:** The authors have declared that no competing interests exist.

# 1. Introduction

Community ambulation requires gait maneuvers to accommodate changes in the environment, such as altering heading to avoid pedestrians or stationary hazards. In such environments, anticipatory control is key for generating locomotor adaptations, and anticipatory gait changes for safe obstacle navigation have been documented [1–3]. However, few have examined anticipatory adjustments in unpredictable, complex walking environments [4–6]. It is generally thought that greater stability during human gait is preferred, and strategies that ensure stability have been described [7]. However, in animal locomotion, some instability is functionally desirable in natural habitats when high maneuverability is important [8, 9]. In humans, anticipatory reduction of stability has been observed in prehension and in standing balance. In prehension behavior, the stability of the force produced by the fingers was reduced when a quick change of the force was expected [10]. In standing balance behavior, the center of pressure was less stable when a postural perturbation was expected [11]. The active reduction of stability of the current motor state is a general feedforward mechanism to facilitate a quick motor change to meet future task demands. It is likely that a similar mechanism, i.e., anticipatory reduction of stability, is used in human gait in challenging environments that require quick maneuvers.

Here, we use the theory of synergies [12, 13] to identify anticipatory changes in the stability of key kinetic variables that determine the body's motion during gait. We quantify stability using synergy indices obtained from uncontrolled manifold analyses of the ground reaction forces and moments [14]. Our premise is that the resultant force and moment acting at the center of mass (CoM), reflecting the inertial effects arising from the current motion of the body segments and the current forces exerted by the muscles, determine future motion via the equations of motion (assuming minimal interference from other external forces). Therefore, stable body movements require consistent resultant forces and moments. Since any change in the motion must be accomplished by changing the underlying kinetics, anticipatory changes should be evident in the kinetics, and particularly in our synergy indices.

In human gait, the whole-body vertical motion is stabilized by a kinetic synergy during the double support phase of unperturbed, steady-state, level walking [15]. That is, the variations in the vertical ground reaction force (GRF) under one foot are compensated by variations in the vertical GRF under the other foot so that the total force is stabilized. We previously extended this result and observed that the resultant force in the medio-lateral (ML) direction and the resultant moment about all three coordinate axes were stabilized by kinetic synergies during the double support phase of curb negotiation [14]. However, it is unknown whether these synergies will be weakened when an unpredictable environment requires quick maneuvers, as having a strong synergy stabilizing the current motion may be counterproductive. The double support phase may be important in this regard: with both feet in contact with the ground, more kinetic degrees of freedom are available, indicating a greater ability to alter the body's movements.

The purpose of the present study was to quantify how synergies during double support are modulated in preparation for meeting environmental demands. We used a foot-targeting task to simulate a challenging environment where the person may have to quickly alter their foot placement for the step immediately *after* stepping down from a curb. We examined the kinetic synergies during curb descent when the expectation of the foot placement for the subsequent step was altered across tasks. Specifically, there was no target for the Baseline task, a stationary target at the preferred foot placement for the Fixed task, and a target that may shift forward or laterally in the Anterior- and Lateral-shift tasks, respectively. We hypothesized that kinetic synergies while stepping down from the curb will be weaker (lower stability) for tasks that

required targeting, compared to a task without a target (H1), similar to what has been observed in manual and postural tasks [10, 11]. We also hypothesized that kinetic synergies will be weakened more when expecting a target shift versus no expectation of target shift, because the increase in uncertainty of the task imposes higher maneuverability demands, and lower stability will help in achieving greater maneuverability (H2). We hypothesized that changes in the kinetic synergies will depend on the direction of the expected target shift (H3). For example, the synergies stabilizing motion in the frontal plane (ML translation and rotation about the anterior-posterior, or AP axis) would be preferentially diminished if the individual expected to alter movement in that plane (e.g., take a wider step). In contrast, synergies stabilizing motion in the sagittal plane (AP translation and rotation about the ML axis) would be preferentially diminished if the individual expected to alter movement in that plane (e.g., take a longer step).

## 2. Methods

### 2.1 Participants

Twenty-seven young healthy adults participated in this study. Data from three participants were excluded due to bad tracking and one participant was excluded because they consistently skipped down the curb, which led to very brief or no double support phase. Data from 23 young participants were used for subsequent analyses (9 males, age 24.0 ± 3.9 years, weight 74.3 ± 19.0 kg, height 1.69 ± 0.10 m, 1 left leg dominant; assessed using the Waterloo Footedness Questionnaire-Revised [16]). Vision was normal or corrected-to-normal. Exclusion criteria were diagnosed neurological diseases or disorders, musculoskeletal injuries, and need for walking aid. All participants provided written informed consent according to the protocols approved by the Institutional Review Board of Purdue University (Protocol number: IRB-2021-52).

### 2.2 Protocol

We instructed participants to wear comfortable walking/athletic shoes that they typically wear when walking outside. We decided to study shod walking because young adults likely spend more time walking with shoes than without, especially over uneven terrain. During the experiment, participants walked in their own shoes on an 8 m walkway with an elevated wooden platform or curb (15 cm high, 1 m wide, and 4 m long). We embedded two force plates in the walkway (AMTI Accugait, MA, USA), one in the curb, and one in the ground (Fig 1A). The plates sampled the ground reaction forces and moments at 1000 Hz. A ten-camera motion capture system (Vicon, Oxford, UK) tracked the body's kinematics at 100 Hz. We placed marker clusters on the lower back and bilateral thigh, shank, and foot segments. We digitized lower-limb joint centers and the toe, heel, and fifth metatarsals of both feet. The force plate and camera data were synchronized using the MotionMonitor software (Innovative Sports Training Inc, IL, USA).

Participants walked at a self-selected speed, stepped down the curb, and continued walking until the end of the walkway. The starting position was identified for each participant during familiarization trials to ensure that participants naturally stepped down the curb with their right foot first. There were four tasks: Baseline, Fixed target, Anterior-shift target, and Lateral-shift target (Fig 1B–1E). These tasks were performed in blocks of 20 or 40 trials; the Baseline task was collected first and the three targeting tasks were block-randomized for each participant. In the Baseline task, participants walked and stepped down the curb with no targets for 20 trials. Average foot placement and step length obtained from these trials determined the target locations for the remaining tasks. The Fixed target task required no anticipation for movement adjustment. In this task, the target was located at the preferred landing position (Fig 1C);

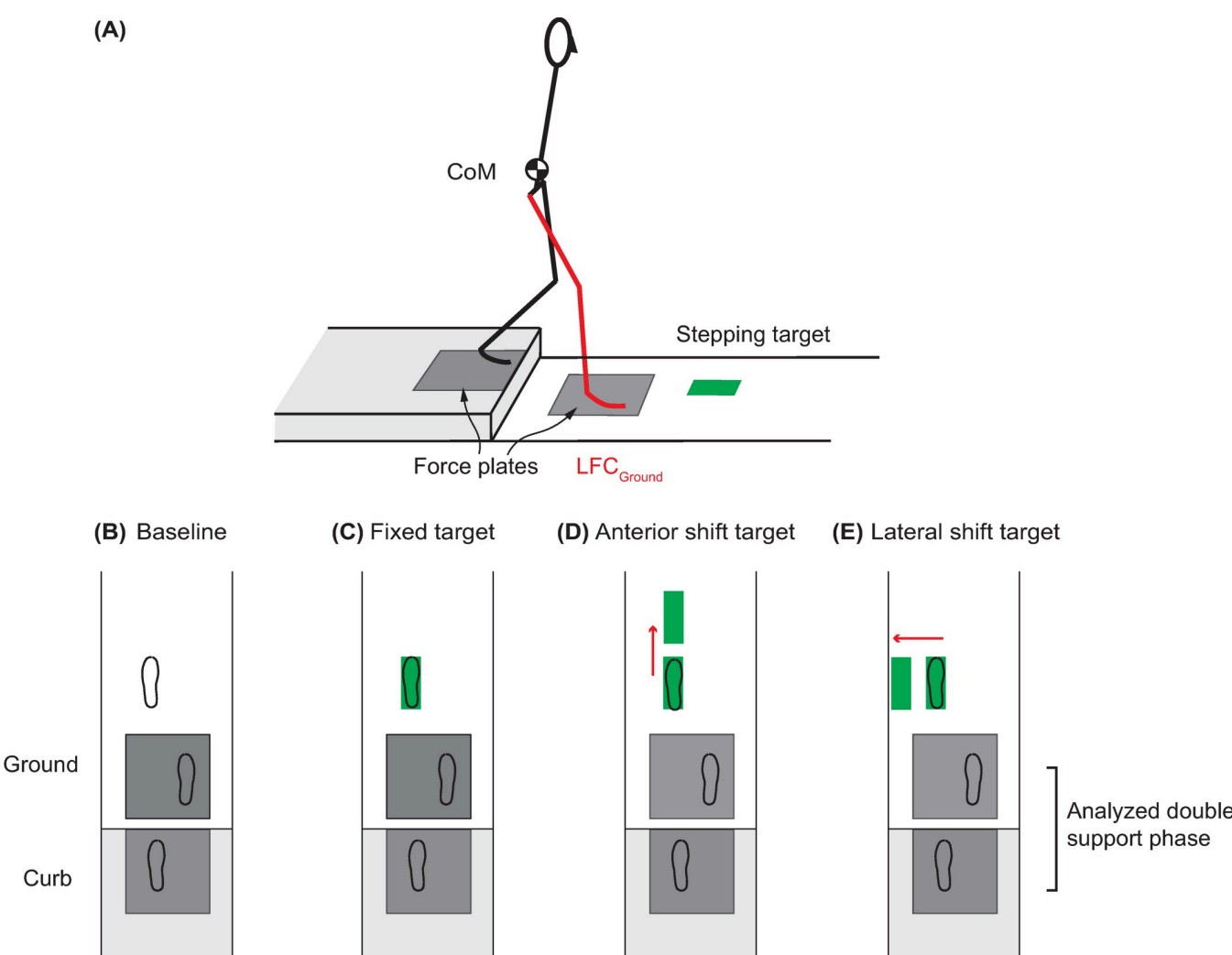

**Fig 1. (A)** Experimental Setup. Two force plates were embedded in the walkway, one in the elevated curb, and one in the ground. A projector was used to present visual stepping target on the walkway. **(B)** Baseline condition. No target was presented. **(C)** Fixed target condition. The target was presented at the preferred foot landing position and remained stationary. **(D, E)** Target shift conditions. A target shift in the anterior **(D)** or lateral **(E)** direction was triggered with 50% probability when the lead foot contacted the ground force plate (LFC$_{Ground}$). Kinetic synergies were quantified at the double support phase while stepping down using the trials where the target did not shift, as illustrated by the footprints in the initial targets in **(D)** and **(E)**.

it was visible and stationary throughout the trial. Participants were instructed to walk down the walkway as before and step on the target as accurately as possible. This task was performed 20 times. The Anterior- and Lateral-shift tasks required anticipating a possible anterior and lateral maneuver, respectively. For both these tasks, the target was visible at the start of each trial and located at the preferred landing position (Fig 1D and 1E). In 50% of the trials, the target shifted to a new position when the lead foot contacted the ground after crossing the curb (LFC$_{Ground}$, Fig 1A). LFC$_{ground}$ was the instant when the vertical GRF exceeded 15 N on the ground force plate. At the beginning of the target shift tasks, participants were told the direction in which the target may shift and were instructed to step on the target as accurately as possible. The forward shift distance was 40% of the average step length [17, 18]. The lateral-shift distance was 20 cm for all participants [17]. The target shift task blocks consisted of 40 trials, with the target shifting for a random set of 20 trials. Participants were given a seated rest break every 20 trials to mitigate fatigue.

The timing of the target shift (at $LFC_{Ground}$, Fig 1A) was set so that the double support phase would include only proactive responses and exclude reactive responses. The longest double support phase in our data across all participants was 193 ms, which is less than the 273 ms reaction time for visual stimuli during walking in young adults [19].

## 2.3 Data analysis

Pelvis center of mass position, toe and heel positions, GRFs, free moments and center of pressure under both feet were obtained from the MotionMonitor software. Kinematic and kinetic data were filtered with a zero-lag, low pass, $4^{th}$-order Butterworth filter with a cut-off frequency of 8 Hz and 20 Hz, respectively.

**Spatial-temporal gait measures.** We quantified the double support duration, gait speed, step length, step width and foot placement locations. Double support phase was the time interval between $LFC_{Ground}$ and the trail foot toe-off from the curb (first instant when vertical GRF on the curb force plate decreased below 15 N). We estimated the whole-body CoM position using the pelvis CoM position. We calculated the instantaneous AP CoM velocity by differentiating the CoM position in the AP direction. Gait speed was the average AP CoM velocity during the double support phase. We used the location of the foot center to quantify foot placement locations for the two steps around the curb and the following target step. The foot center locations were determined as the midpoint of the digitized heel and toe positions when the foot was flat. The foot flat instant was 300 ms after the corresponding heel contacted the ground. Heel contact was identified using the AP heel position [20]. We visually inspected each trial and ensured that the foot was flat at these time instants.

**Target stepping performance.** The stepping error for the targeting tasks was the length of the vector between the foot center and the target center in the horizontal plane (the vertical coordinate of the foot center was ignored). We quantified the root mean squared error (RMSE) of the vector lengths for each task. We quantified the foot placement RMSE separately for the target-shift trials and the no-shift trials for the Anterior and Lateral-shift tasks. For the Baseline task, we computed the standard deviation in the foot center positions.

**Uncontrolled manifold (UCM) analysis.** In general, the UCM analysis partitions the across-trial variance in the redundant set of input variables into a component that maintains the performance variable ($V_{UCM}$) and another orthogonal component that changes the performance variables ($V_{ORT}$) [13]. A synergy index is obtained as the normalized excess of the $V_{UCM}$ over $V_{ORT}$.

We performed the UCM analysis to investigate whether the across-trial variance in the GRF and free moments under the two feet (called the input variables) is structured to maintain the resultant forces and moments (called the performance variables) during the double support phase. We considered the vertical, AP and ML components of resultant force and moments as separate performance variables, and therefore, performed six separate UCM analyses (see S1 Appendix for more details). A resultant force component was determined by the GRFs under both feet during double support. A resultant moment component was determined by the GRFs (and free moment for the resultant moment about vertical axis [14]) under both feet. At every 1% of the double support phase, we quantified the synergy index ($\Delta V$) and its z-transformed value ($\Delta Vz$) for each performance variable. Finally, we averaged the $\Delta Vz$ values over the double support phase to obtain a single synergy index for each performance variable and for each task. Each UCM analysis is associated with a discriminating value $\Delta Vz^*$; Table 2. A $\Delta Vz$ greater than the corresponding discriminating value indicates existence of a synergy. A greater $\Delta Vz$ value indicates a stronger synergy, and vice-versa.

**Trials included in the data analysis.** Stepping performance was quantified for six different sets of trials: Baseline, Fixed target, targets that may shift anteriorly but did not, targets that shifted anteriorly, targets that may shift laterally but did not, and targets that shifted laterally. Spatiotemporal and UCM measures were quantified for four of these six sets of trials; the two sets of trials where the targets shifted were not used. Thus, spatiotemporal and UCM measures all had the same foot placement, but the expectation changed across the four conditions. Any differences in these measures are attributable only to the expectation of a maneuver.

## 2.4 Statistical analysis

To determine if participants adhered to the targeting task, we performed one-way ANOVA (6 levels: Baseline, Fixed target, no shift and shift trials from the Anterior-shift task, no shift and shift trials from the Lateral-shift task) on the RMSE. To determine the presence of anticipatory locomotor adjustments, we performed one-way ANOVAs (4 Tasks: Baseline, Fixed target, Anterior-shift target, Lateral-shift target) on traditional gait measures (gait speed, double support duration, lead and trail foot placement locations in AP and ML directions, step length and step width).

To determine the presence of synergies, we conducted separate two-tailed t-tests for comparing $\Delta Vz$ to the appropriate discriminating value ($\Delta Vz^*$; Table 2) for each performance variable and each task. To determine if the synergies were modulated with tasks, we performed the same ANOVA separately on the six synergy indices using the GLIMMIX procedure, with participants as the random effect. We performed all pair-wise post hoc comparisons with Tukey-Kramer adjustments when significant effect of Task was observed. Analyses were performed in SAS (Cary, NC, USA). The level of significance was set at 0.05.

## 3. Results

### 3.1 Targeting performance

RMSE was significantly different across tasks (Table 1, Fig 2). Post hoc comparisons revealed that all target tasks had smaller RMSE compared to Baseline, showing that participants performed the targeting task as instructed. The shift trials in Lateral-shift task had larger RMSE compared to the other target tasks.

### 3.2 Spatial-temporal gait measures

Significant effect of Task was observed for gait speed (Table 1, Fig 3A). Post hoc comparisons showed that gait speed significantly reduced for the Lateral-shift task compared to the

**Table 1. Spatial-temporal gait parameters and associated ANOVA results.**

| Target stepping accuracy | | $F_{(5,110)}$ | p-value | $\eta_p^2$ |
|---|---|---|---|---|
| RMSE | | 36.34 | <0.001 | 0.62 |
| **Gait measures** | | $F_{(3,66)}$ | **p-value** | $\eta_p^2$ |
| Gait speed | | 8.46 | **<0.001** | 0.28 |
| Double support phase | | 2.06 | 0.114 | – |
| Trail foot placement | AP | 3.56 | **0.019** | 0.14 |
| | ML | 4.92 | **0.004** | 0.18 |
| Lead foot placement | AP | 48.36 | **<0.001** | 0.19 |
| | ML | 6.05 | **0.001** | 0.22 |
| Step length | | 44.54 | **<0.001** | 0.67 |
| Step width | | 2.52 | 0.066 | – |

Significant effects of Task are bolded.

**Table 2. One-sample t-tests to determine whether the synergy index (ΔVz) is different from the corresponding discriminating value (ΔVz*) for the three resultant force and the three resultant moment components for each task.**

| Synergy | Axis | ΔVz* | Task | mean | $t_{(22)}$ | p-value | Cohen's d |
|---|---|---|---|---|---|---|---|
| Resultant force synergy | AP | 0 | Baseline | -0.07 | -1.91 | 0.069 | – |
| | | | Fixed target | -0.07 | -2.03 | 0.055 | – |
| | | | Anterior | -0.07 | -1.91 | 0.069 | – |
| | | | Lateral | -0.07 | -3.10 | **0.005** | -0.65 |
| | ML | 0 | Baseline | 0.12 | 3.60 | **0.002** | 0.75 |
| | | | Fixed target | 0.09 | 4.38 | **<0.001** | 0.91 |
| | | | Anterior | 0.11 | 3.73 | **0.001** | 0.78 |
| | | | Lateral | 0.09 | 2.99 | **0.007** | 0.62 |
| | Vertical | 0 | Baseline | 0.06 | 1.78 | 0.089 | – |
| | | | Fixed target | 0.08 | 3.62 | **0.002** | 0.75 |
| | | | Anterior | 0.07 | 1.90 | 0.071 | – |
| | | | Lateral | 0.08 | 2.02 | 0.056 | – |
| Resultant moment synergy | AP | 0.55 | Baseline | 2.04 | 29.67 | **<0.001** | 6.19 |
| | | | Fixed target | 1.89 | 35.27 | **<0.001** | 7.35 |
| | | | Anterior | 1.86 | 32.90 | **<0.001** | 6.86 |
| | | | Lateral | 1.86 | 33.62 | **<0.001** | 7.01 |
| | ML | 0.55 | Baseline | 1.66 | 25.02 | **<0.001** | 5.22 |
| | | | Fixed target | 1.58 | 19.38 | **<0.001** | 4.04 |
| | | | Anterior | 1.54 | 17.00 | **<0.001** | 3.55 |
| | | | Lateral | 1.53 | 16.82 | **<0.001** | 3.51 |
| | Vertical | 0.80 | Baseline | 2.39 | 39.67 | **<0.001** | 8.27 |
| | | | Fixed target | 2.34 | 41.36 | **<0.001** | 8.63 |
| | | | Anterior | 2.35 | 34.44 | **<0.001** | 7.18 |
| | | | Lateral | 2.36 | 51.81 | **<0.001** | 10.80 |

other three tasks. No effect of Task was observed for the double support duration (Table 1, Fig 3B).

Even though the target remained at the preferred foot location for the analyzed trials in all targeting tasks, a significant effect of Task was observed for AP and ML foot placements around the curb (Table 1). Post hoc comparisons revealed that the trail foot (on the curb) was placed closer to the curb edge and more lateral (left) for the Lateral-shift task compared to the Baseline task (Fig 4A). The lead foot (on the ground) was placed more forward for the Fixed target task compared to the Baseline task, and more forward for the Anterior- and Lateral-shift tasks compared to Fixed target task. The lead foot was also placed more lateral (right) for the Fixed target and Anterior-shift tasks compared to the Baseline task. As a result, step length was increased for the targeting tasks compared to Baseline, and it increased the most for the Anterior-shift task. Step width did not change across tasks (Table 1).

### 3.3 Synergies for the resultant force

The separate t-tests revealed that the synergy index for the resultant ML force was significantly greater than zero for all tasks. The Synergy index for the resultant AP force was significantly lower than zero for the Lateral-shift task, and it was not significantly different from zero for the other tasks. The synergy index for the resultant vertical force was significantly greater than zero for the Fixed target task, and it was not significantly different from zero for the other tasks (Table 2).

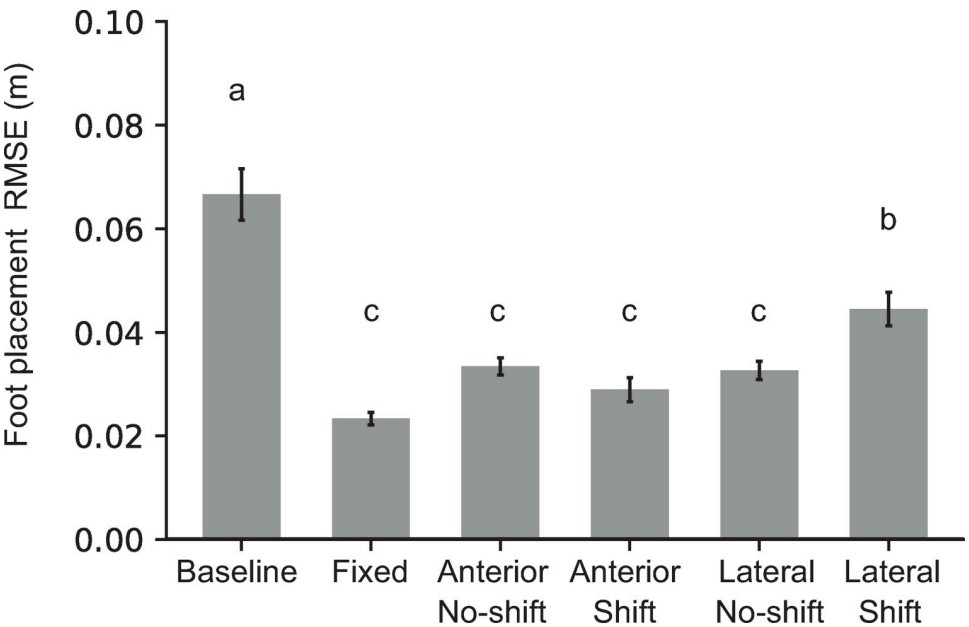

**Fig 2. Mean and standard error for foot placement RMSE for each task.** RMSE is computed separately for the no-shift and shift trials for the Anterior-shift and Lateral-shift tasks. Means with different letters are significantly different from one another.

None of the ANOVAs on the synergy indices for the resultant force components revealed a significant effect of Task (Fig 5; Table 3).

### 3.4 Synergies for the resultant moment

The separate t-tests revealed that the synergy index for the resultant moments along all three directions was significantly greater than the corresponding $\Delta Vz^*$ for all tasks (Table 2).

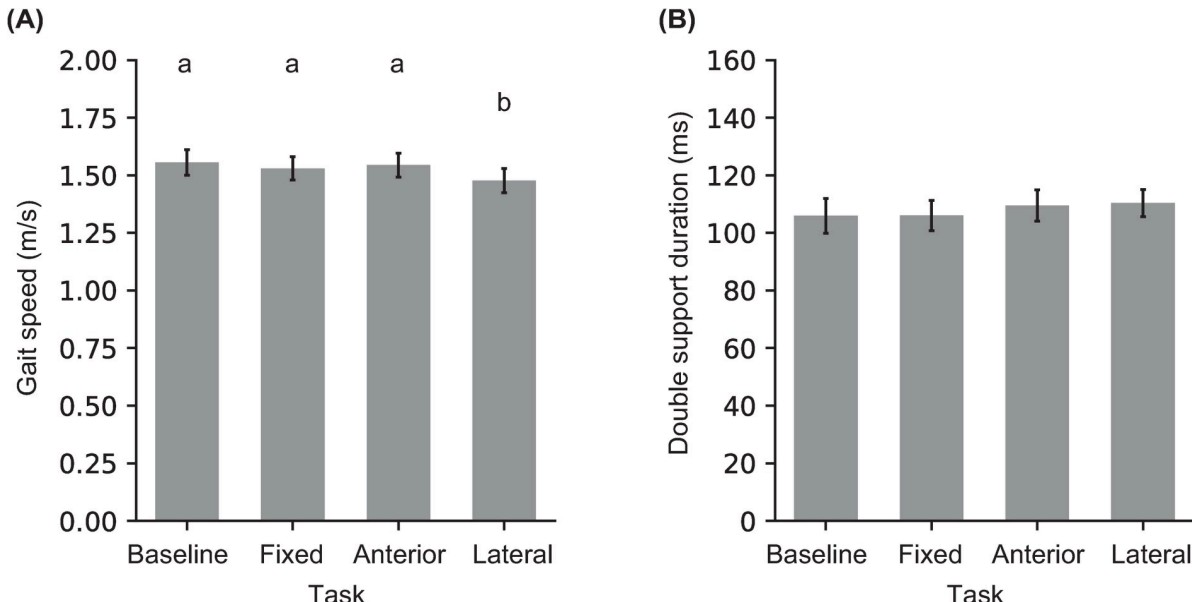

**Fig 3.** **(A)** Mean and standard error for gait speed during double support phase, and **(B)** double support duration for the four tasks. Means with different letters are significantly different from one another.

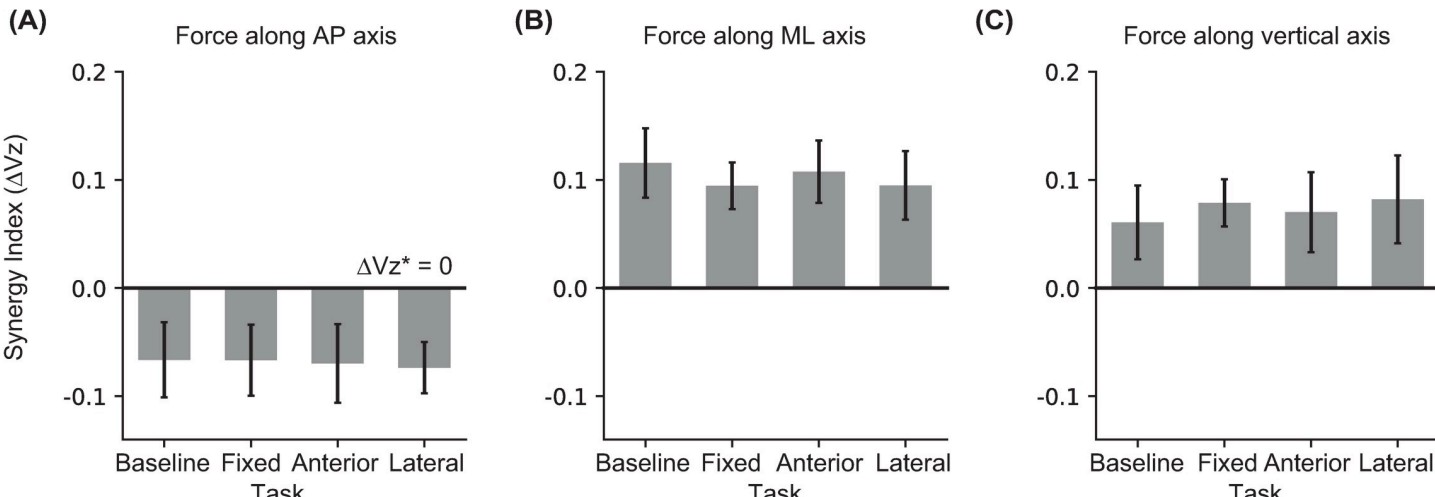

**Fig 4. (A)** Mean and SE of foot placement locations relative to the curb edge (horizontal line) and the midpoint of the walkway for the four tasks. Error bars represent across-participant standard error. The dashed footprints are a zoomed-in view of the two foot placement locations with a common scale for provided on the right. **(B, C)** Mean and standard error for step length and step width. Means with different letters are significantly different from one another.

**Fig 5.** The mean and SE for synergy index (ΔVz) for stabilizing resultant force along the **(A)** AP, **(B)** ML and **(C)** vertical axis during the double support phase for the four tasks. The horizontal lines in each plot show the discriminating values for each synergy index.

**Table 3. ANOVA results for the six synergy measures.**

| Synergy | Axis | $F_{(3,66)}$ | p-value | $\eta_p^2$ |
|---|---|---|---|---|
| Resultant force synergy | AP | 0.01 | 0.998 | – |
| | ML | 0.15 | 0.926 | – |
| | Vertical | 0.11 | 0.953 | – |
| Resultant moment synergy | AP | 6.64 | **0.001** | 0.23 |
| | ML | 1.80 | 0.156 | – |
| | Vertical | 0.56 | 0.641 | – |

Significant effects of Task are bolded.

The ANOVA analyses revealed a significant effect of Task for the synergy stabilizing the resultant moment about the AP axis. Post hoc comparisons revealed that the synergy index decreased for all targeting tasks compared to Baseline. The ANOVAs on the synergy indices for resultant moments about the other two axes did not reveal an effect of Task (Fig 6; Table 3).

## 4. Discussion

In the present study, we investigated the effects of future maneuverability demands on the stability ensured by the covariation between the GRFs (and free moments) under the two feet during the double-support phase of stepping down from a curb. We extended the current evaluation of kinetic locomotor synergies by examining the modulation of these synergies in a complex environment with maneuverability demands. Our hypothesis that kinetic synergies will be reduced when there is subsequent precision stepping demand was partially supported. Our hypotheses that (1) kinetic synergies will be reduced when expecting the target to shift versus a stationary target, and (2) synergies will be reduced preferentially depending on the direction of the expected maneuver were not supported.

The mechanics and control of step negotiation are well explored. However, the double support phase, with the feet planted on surfaces at different heights, is understudied, consistent

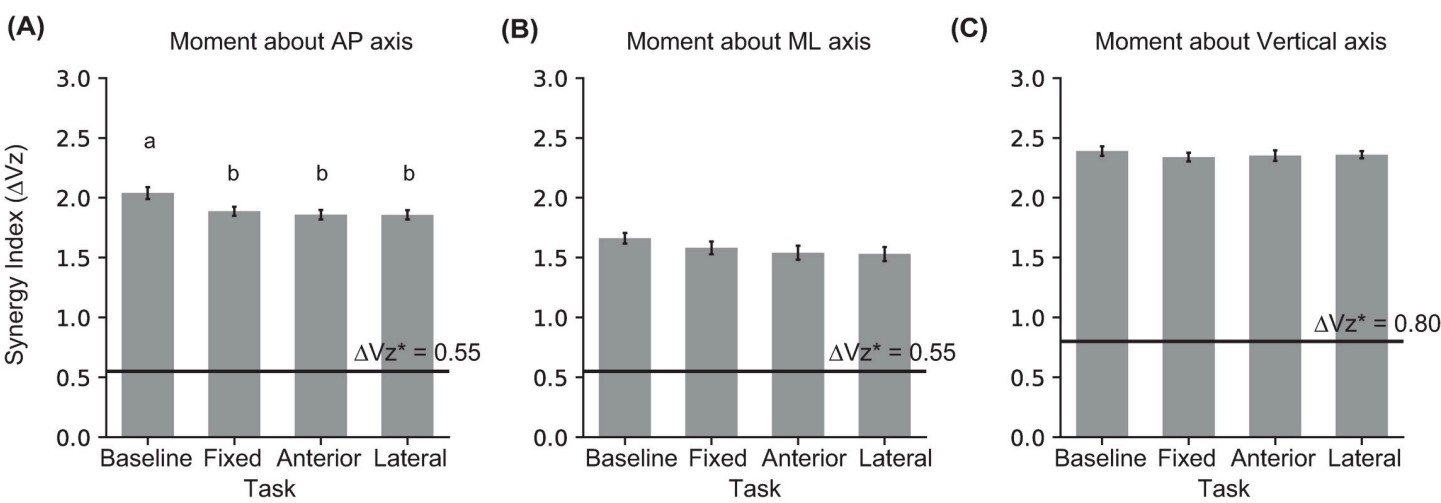

**Fig 6.** Mean and standard error for the synergy index (ΔVz) for stabilizing resultant moment about the **(A)** AP, **(B)** ML, and **(C)** vertical axes during the double support phase. The horizontal lines in each plot show the discriminating values for each synergy index. ΔVz greater than the corresponding discriminating value indicates presence of synergy. Means with different letters are significantly different from one another.

with a relative neglect of this phase in the locomotion literature [21]. When both feet contact the ground, there are more kinetic degrees of freedom, which offers greater control authority over the gait cycle [14, 22]. Therefore, this phase of gait, although brief (lasting 20% of the gait cycle), may be an opportunity for modulating the body's motion for achieving specific kinematic goals to navigate hazards. The main finding of this study contradicted our expectations: the kinetic synergies during double support of curb descent were minimally influenced by expected subsequent maneuvers. Nevertheless, our results corroborate our earlier findings, validating the novel synergy analysis that we introduced, and highlight the robustness of the ground reaction variable synergies engaged in stabilizing whole-body angular motions during the double support phase of curb descent [14]. We discuss the implications of this finding below.

### 4.1 Synergies stabilizing resultant forces

The variance structure in the GRFs is quite robust; the experimental manipulations had minimal effect on the synergy indices for the resultant forces. The index for the resultant ML force was positive, and it was not affected by the upcoming targeting requirements. This result is compatible with previous work indicating active control of gait in the ML direction [23, 24]. The vertical and AP forces, on the other hand, showed no synergistic co-variation for most cases (Table 2), corroborating our earlier findings for the Baseline task [14]. The behavior of these forces can be understood based on their roles in stabilizing linear and rotational motion.

During double support, each GRF pair contributes to a resultant force and resultant moments about two axes (Table 4). These contributions are evident from the statics analysis of the body posture in double support, i.e., from the spatial relationships between the GRFs lines of action and the CoM location [14]. For example, the vertical GRFs determine the resultant vertical force at the CoM, but also contribute to the resultant moment about the ML axis. Critically, attempting to stabilize the resultant vertical force will impair the stabilization of resultant ML moment, since the forces must covary *negatively* to stabilize the resultant vertical force, but they must covary *positively* to stabilize the resultant ML moment.

Therefore, the lack of synergy in resultant vertical and AP forces indicates that the angular motion in sagittal plane is stabilized at the cost of linear motion in the vertical and AP directions, likely to diminish the risk of a forward fall during curb descent. This interpretation is further supported by a consistent pattern in the data across three studies. The stabilization of the resultant AP and vertical forces diminishes across tasks as the person gains, maintains or loses height: both resultant forces are stabilized for curb ascent [14], only the resultant vertical force is stabilized during level walking [15], and neither resultant force is stabilized while stepping down. The risk and the consequences of a forward fall are higher when stepping down. Therefore, controlling forward pitching motion about the ML axis becomes more important, and the vertical and AP forces are engaged in stabilizing the resultant moment about the ML axis instead of their respective sums, reflecting a shift in control objectives in response to the task.

**Table 4. Directions of covariations in the force pairs required to stabilize each performance variable.**

| Force pair | Performance variable | | | | | |
|---|---|---|---|---|---|---|
| | $F_{Resultant-ML}$ | $F_{Resultant-AP}$ | $F_{Resultant-V}$ | $M_{Resultant-ML}$ | $M_{Resultant-AP}$ | $M_{Resultant-V}$ |
| ML | Negative | | | | Negative | Positive |
| AP | | Negative | | Negative | | Positive |
| Vertical | | | Negative | Positive | Positive | |

## 4.2 Synergies stabilizing resultant moments

The synergy indices for resultant moments about all three axes were higher than the corresponding discriminating values (with large effect sizes; Table 2), corroborating our earlier findings for the Baseline task [14], and extending them to more complex tasks. Subsequent stepping requirements reduced the synergy index for the resultant moment about the AP axis by 12%, i.e., the angular motion in the frontal plane became less stable. In contrast to our expectations, this change was insensitive to the nature of the upcoming task; the decline was consistent when the target was fixed, and when the target was likely to jump in the AP or the ML direction.

The decline in this synergy index could arise from increased variability in the trailing ankle push-off for the targeting tasks. Modulating ankle push-off is a known strategy for maintaining balance during locomotion [25, 26]; modulating ankle contributions may be an effective mechanism for modulating synergies during double support as well. Ankle push-off affects not only AP linear motion, but also angular momentum about the AP and ML axes [27]. In the target shift tasks, modulating ankle push-off may be used to increase step length or step width when needed. Furthermore, for the Fixed target task, the presence of the target constrains both step length and width, and this may also influence ankle push-off, and hence the synergy index. Foot target requirements–even when targets are located at preferred foot locations–are known to affect kinematic locomotor synergies [28]. The speculations regarding the association between ankle push-off and kinetic GRF synergies needs to be tested. Additionally, other mechanisms that could result in a lower synergy index, such as hip torque contributions, need to be explored.

The decline in the synergy index indicates active control of the stability of angular motion in the frontal plane, and the lack of Task effect on other synergy indices suggests that the same level of control was not employed for other resultant forces and moments. It is known that human gait requires more active control along the ML compared to the AP direction, and much work has focused on identifying the mechanisms for ML control [23, 25, 26]. However, this literature focuses mainly on ML linear motion, and most quantitative measures that assess ML balance do not consider the angular motion of the body [29]. Although whole-body angular momentum has been quantified during locomotion [27, 30–32], our work shows that kinetic synergy analysis can improve our understanding of how we modulate body rotations in challenging environments. Furthermore, previous work on identifying the direction-specificity of locomotor control has focused on reactive foot placement modulations [33–35]; our work extends these ideas to bilateral GRF coordination during double support, and critically, from reactive to proactive modulations for future maneuvers. However, we acknowledge that the insensitivity of the synergy adjustment to the nature of the upcoming targeting task requires further study.

## 4.3 Robust kinetic synergies during curb descent

Overall, the kinetic synergies during double support of curb descent were robust and minimally influenced by expected maneuvers, with only one of the six synergy indices showing a Task effect. There are two possible explanations for this finding. First, the consequences of failure at the two tasks–stepping down from the curb and subsequent targeting–are not equivalent. Loss of balance is more likely during curb descent, and a fall would have a severe cost. In comparison, failure to step into the visual target has minimal consequences. The main function of the synergies, then, is to maintain angular stability (reflected by the high moment synergies, and low force synergies), and this prerogative was not overcome by the upcoming targeting tasks that we used. This interpretation is compatible with the observation that angular

momentum is tightly regulated while walking down stairs or inclines [31, 32]. In fact, our work here and earlier [14] extends this view to moments about all three coordinate axes.

Secondly, participants had additional opportunities to alter their movements and accomplish the targeting task. Slower walking speed for the Lateral-shift task reflected a strategy to wait to see if the target would shift. More forward ground foot placement (11% increase in the distance from the curb edge relative to Baseline) and longer step length (6% increase relative to Baseline) may reflect a strategy where the participant biased earlier foot placements in the direction of the potential target shift to facilitate targeting. Furthermore, although we did not quantify the swing phase, previous research has shown that young adults make reactive adjustments to track a moving foot placement target as late as mid-swing [36, 37]. Given these anticipatory and reactive opportunities to alter foot placement, participants may have preserved the kinetic synergies during double support to offset the higher risk of loss of balance during curb descent.

### 4.4 Limitations

The tasks we selected to simulate a challenging environment may not have sufficiently challenged young healthy participants. It is possible that if the task were more challenging–such as the target shifting in multiple directions rather than one direction only–we would have observed decline in the synergy indices. Thus, we should use caution when describing the synergies as robust until further environments are considered.

## 5. Conclusion

We have demonstrated that the kinetic synergies in the ground reaction forces and moments during the double support phase of curb descent are robust to and minimally influenced by expected foot targeting maneuvers that we used. With both feet contacting the ground during double support, there are additional kinetic degrees of freedom with which to affect the motion of the body. The main function of these kinetic variables is to maintain rotational stability (reflected by the high resultant moment synergies, and low resultant force synergies), and this prerogative was minimally influenced by the subsequent targeting tasks that we used, likely because the cost of losing balance while descending the curb would be higher than the cost of a misstep on a visual target. The synergy stabilizing rotational motion in the frontal plane declined when subsequent targeting was required. Further work is required to identify why the synergy adjustment was independent of the direction of target shift.

## Supporting information

**S1 Appendix.**
(DOCX)

## Author Contributions

**Conceptualization:** Satyajit Ambike.

**Data curation:** Chuyi Cui, Ashwini Kulkarni.

**Formal analysis:** Chuyi Cui, Satyajit Ambike.

**Funding acquisition:** Chuyi Cui, Satyajit Ambike.

**Investigation:** Chuyi Cui, Ashwini Kulkarni, Shirley Rietdyk, Satyajit Ambike.

**Methodology:** Chuyi Cui, Shirley Rietdyk, Satyajit Ambike.

**Project administration:** Shirley Rietdyk, Satyajit Ambike.

**Resources:** Satyajit Ambike.

**Software:** Chuyi Cui.

**Supervision:** Shirley Rietdyk, Satyajit Ambike.

**Validation:** Chuyi Cui, Shirley Rietdyk, Satyajit Ambike.

**Visualization:** Chuyi Cui, Shirley Rietdyk, Satyajit Ambike.

**Writing – original draft:** Chuyi Cui, Ashwini Kulkarni, Shirley Rietdyk, Satyajit Ambike.

**Writing – review & editing:** Chuyi Cui, Ashwini Kulkarni, Shirley Rietdyk, Satyajit Ambike.

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
