## [Decision Letter · Decision Letter 0]

2 Sep 2022

PONE-D-22-11648Locomotion control during curb descent: Bilateral ground reaction variables covary consistently during the double support phase regardless of future foot placement constraintsPLOS ONE

Dear Dr. Ambike,

Thank you for submitting your manuscript to PLOS ONE. After careful consideration, we feel that it has merit but does not fully meet PLOS ONE’s publication criteria as it currently stands. Therefore, we invite you to submit a revised version of the manuscript that addresses the points raised during the review process.

Please see the reviewers' comments below. Please note that while Reviewer #2 was quite critical regarding the overall significance of this study compared to previous work in the field, and we respect that opinion, it is our assessment that this does not fail our criteria for publication; novelty (in the sense of significance or impact) is not a requirement for publication in PLOS ONE for original research that otherwise meets our requirements for quality of execution, reporting, and ethical standards. As such, while you may wish to clarify the specific contributions this work brings to the field, we require you only to address the other comments raised by each reviewer.

We look forward to receiving your revised manuscript.

Kind regards,

Hugh Cowley

Staff Editor

PLOS ONE

Journal Requirements:

Reviewers' comments:

Reviewer's Responses to Questions

**Comments to the Author**

1. Is the manuscript technically sound, and do the data support the conclusions?

Reviewer #1: Yes

Reviewer #2: Yes

2. Has the statistical analysis been performed appropriately and rigorously? 

Reviewer #1: Yes

Reviewer #2: Yes

3. Have the authors made all data underlying the findings in their manuscript fully available?

Reviewer #1: Yes

Reviewer #2: Yes

4. Is the manuscript presented in an intelligible fashion and written in standard English?

Reviewer #1: Yes

Reviewer #2: Yes

5. Review Comments to the Author

Reviewer #1: Thank you for this well-written and interesting paper. Also, your description of UCM was well done-- terse but complete.

I have two comments for things that you need to address before publication: 1) where is the "foot center" line 170 p8 and throughout. Specifically, how do you define the foot center? I can imagine that the target center is the geometric center of the box. Obviously, this is a critical component of your results. 2) Were you participants shod or unshod? Please specify and if shod, discuss what kind of shoes and how that might have effected the results.

Reviewer #2: The paper is an extension of the “synergy index” line of work on which Dr. Ambike has developed and published extensively. The current focus is whether a targeted stepping requirement subsequent to a step up or down on a curb influence the synergy indices during the “curb step” compared to a condition in which no targeted stepping is required. The results showed minor effects, primarily in gait parameters, of the targeted stepping task. The authors conclude that performance on the targeting task had “low cost” relative to the task of maintaining upright stability while stepping up or down. Therefore, the targeting task had little effect on the synergy indices.

I agree with the authors’ interpretation of the results. There were small adjustments in gait parameters during the targeting conditions compared to baseline. But clearly, the targeting tasks were not demanding enough to threaten the stability of the upright body while stepping up or down.

From this perspective, the paper reproduces many of the results found in previous work, such as synergy indices related to free moments are related to upright stability moreso than ground reaction forces. The attempt to modify this essential finding was not successful. Such results contribute to the synergy index line of work in showing the nervous system prioritizes constraints while moving through the environment. Not a dramatic contribution.

Minor Comments

1. P 10 1st PP confusing sentences - please clarify. “All remaining measures included only the trials where the target did not shift. Thus, in the remaining measures, the foot placements during curb descent as well as the next step were the same in every task, and the expectation of a target shift differs across the tasks.”

2. P 10, line 221 – “…the same the ANOVA…” remove “the” before ANOVA.

3. Figure 2 caption – What are “…dissimilar letters…”?

6. PLOS authors have the option to publish the peer review history of their article (what does this mean?). If published, this will include your full peer review and any attached files.

Reviewer #1: No

Reviewer #2: No

---

## [Author Response · Author response to Decision Letter 0]

15 Sep 2022

We thank the reviewers for taking the time to review our manuscript and for their valuable feedback. Below is our point-by-point response to the editor’s comments and then the reviewers’ comments.

Editor’s comment:

Please note that while Reviewer #2 was quite critical regarding the overall significance of this study compared to previous work in the field, and we respect that opinion, it is our assessment that this does not fail our criteria for publication; novelty (in the sense of significance or impact) is not a requirement for publication in PLOS ONE for original research that otherwise meets our requirements for quality of execution, reporting, and ethical standards. As such, while you may wish to clarify the specific contributions this work brings to the field, we require you only to address the other comments raised by each reviewer.

RESPONSE: Thank you for the comment. We respect this comment by Reviewer 2. We have responded to it in this document below. 

The main contributions of this work are: (1) the robust nature of the synergies, and (2) the replication of the synergy results from our previous work (Cui et al 2018). We think that the point about robust synergies is quite salient – in the title, abstract, discussion and conclusion sections. We adjusted some language to make the point about replication more prominent (Lines 328 - 333). 

Comments by Reviewer 1: 

Thank you for this well-written and interesting paper. Also, your description of UCM was well done-- terse but complete.

RESPONSE: We thank the reviewer for the encouragement.

I have two comments for things that you need to address before publication: 

1) where is the "foot center" line 170 p8 and throughout. Specifically, how do you define the foot center? I can imagine that the target center is the geometric center of the box. Obviously, this is a critical component of your results. 

RESPONSE: Foot center was defined as the average (geometric center) of the digitized toe and heel positions. The foot center was computed only when the foot was fully on the ground. For computing stepping error, the vertical coordinate of the foot center was ignored and only the AP and ML coordinates were utilized. We added text (Lines 173 and 181) to clarify this point. 

2) Were you participants shod or unshod? Please specify and if shod, discuss what kind of shoes and how that might have affected the results.

RESPONSE: All participants walked with their own shoes. We instructed the participants to wear comfortable athletic shoes to the experiment. Walking in their own shoes adds to the ecological validity of the gait task. Compared to walking over level ground, there is greater impact on the foot while stepping down from a curb, and this may influence behavior. We decided to study shod walking because young adults likely spend more time walking with shoes than without, especially over uneven terrain. 

We added text in the Methods section (Line 111) to clarify this point. 

As for the effect of performing our tasks with and without shoes: There would likely only be minor differences across the shod/no-shod condition, if any. Specific values of the outcomes might change, but we expect the same significant effects in our statistical comparisons. There is no good reason why the synergy indices – the focus of this work – would change differently for expectation of shifting targets when wearing shoes versus barefoot. 

Comments by Reviewer 2: 

The paper is an extension of the “synergy index” line of work on which Dr. Ambike has developed and published extensively. The current focus is whether a targeted stepping requirement subsequent to a step up or down on a curb influence the synergy indices during the “curb step” compared to a condition in which no targeted stepping is required. The results showed minor effects, primarily in gait parameters, of the targeted stepping task. The authors conclude that performance on the targeting task had “low cost” relative to the task of maintaining upright stability while stepping up or down. Therefore, the targeting task had little effect on the synergy indices.

I agree with the authors’ interpretation of the results. There were small adjustments in gait parameters during the targeting conditions compared to baseline. But clearly, the targeting tasks were not demanding enough to threaten the stability of the upright body while stepping up or down.

From this perspective, the paper reproduces many of the results found in previous work, such as synergy indices related to free moments are related to upright stability more so than ground reaction forces. The attempt to modify this essential finding was not successful. Such results contribute to the synergy index line of work in showing the nervous system prioritizes constraints while moving through the environment. Not a dramatic contribution.

RESPONSE: We respectfully suggest that although not dramatic, the contributions of this work are significant. We show that, at least to the set of expected maneuvers that we used, these synergies are robust. This was not known before. Furthermore, our earlier paper was the first to introduce the UCM analysis of all ground reaction variables. Given the novelty of this analysis, it is quite important to demonstrate the replicability of the results, which we have achieved here. 

Minor Comments

1. P 10 1st PP confusing sentences - please clarify. “All remaining measures included only the trials where the target did not shift. Thus, in the remaining measures, the foot placements during curb descent as well as the next step were the same in every task, and the expectation of a target shift differs across the tasks.” 

RESPONSE: By analyzing only the trials where the target did not shift in the Anterior- and Lateral-shift conditions, we are certain that any difference in the outcomes is attributable to only to the expectation of a maneuver. We have expanded the section on how the data were analyzed to improve clarity (Lines 208-214). 

2. P 10, line 221 – “…the same the ANOVA…” remove “the” before ANOVA.

RESPONSE: Thank you for pointing out the typo. We edited the text as suggested (Line 227).

3. Figure 2 caption – What are “…dissimilar letters…”?

RESPONSE: We replaced “dissimilar letters” with “…different letters…” in all the captions (Lines 564, Line 567, Line 574, and Line 585)

---

## [Editor Report · Decision Letter 1]

19 Sep 2022

Locomotion control during curb descent: Bilateral ground reaction variables covary consistently during the double support phase regardless of future foot placement constraints

PONE-D-22-11648R1

Dear Dr. Ambike,

We’re pleased to inform you that your manuscript has been judged scientifically suitable for publication and will be formally accepted for publication once it meets all outstanding technical requirements.

Kind regards,

Hugh Cowley

Staff Editor

PLOS ONE
---

## [Editor Report · Acceptance letter]

26 Sep 2022

PONE-D-22-11648R1 

Locomotion control during curb descent: Bilateral ground reaction variables covary consistently during the double support phase regardless of future foot placement constraints 

Dear Dr. Ambike:

I'm pleased to inform you that your manuscript has been deemed suitable for publication in PLOS ONE. Congratulations! Your manuscript is now with our production department. 

Kind regards, 

on behalf of

Mr Hugh Cowley 

Staff Editor

PLOS ONE